# Preparation and Characterization of Extracellular Matrix Hydrogels Derived from Acellular Cartilage Tissue

**DOI:** 10.3390/jfb13040279

**Published:** 2022-12-07

**Authors:** Tsong-Hann Yu, Tsu-Te Yeh, Chen-Ying Su, Ni-Yin Yu, I-Cheng Chen, Hsu-Wei Fang

**Affiliations:** 1Department of Orthopedics, Tri-Service General Hospital, National Defense Medical Center, No. 325, Sec. 2, Chenggong Rd., Taipei 114202, Taiwan; 2Department of Chemical Engineering and Biotechnology, National Taipei University of Technology, No. 1, Sec. 3, Zhongxiao E. Rd., Taipei 10608, Taiwan; 3Accelerator for Happiness and Health Industry, National Taipei University of Technology, No. 1, Sec. 3, Zhongxiao E. Rd., Taipei 10608, Taiwan; 4Institute of Biomedical Engineering and Nanomedicine, National Health Research Institutes, No. 35, Keyan Road, Zhunan 35053, Taiwan

**Keywords:** acellular extracellular matrix, hydrogel, decellularization, scaffold, tissue adhesion, tissue engineering, biomaterials

## Abstract

Decellularized matrices can effectively reduce severe immune rejection with their cells and eliminated nucleic acid material and provide specific environments for tissue repair or tissue regeneration. In this study, we prepared acellular cartilage matrix (ACM) powder through the decellularization method and developed ACM hydrogels by physical, chemical, and enzymatic digestion methods. The results demonstrated that the small size group of ACM hydrogels exhibited better gel conditions when the concentration of ACM hydrogels was 30 and 20 mg/mL in 1N HCl through parameter adjustment. The data also confirmed that the ACM hydrogels retained the main components of cartilage: 61.18% of glycosaminoglycan (GAG) and 78.29% of collagen, with 99.61% of its DNA removed compared to samples without the decellularization procedure (set as 100%). Through turbidimetric gelation kinetics, hydrogel rheological property analysis, and hydrogel tissue physical property testing, this study also revealed that increasing hydrogel concentration is helpful for gelation. Besides, the ex vivo test confirmed that a higher concentration of ACM hydrogels had good adhesive properties and could fill in cartilage defects adequately. This study offers useful information for developing and manufacturing ACM hydrogels to serve as potential alternative scaffolds for future cartilage defect treatment.

## 1. Introduction

Cartilage defects are one of the most prevalent signs of osteoarthritis—a degenerative disease that affects millions of people globally and imposes a considerable socio-economic burden on society [1]. Although articular cartilage allows for the relative mobility of opposing joint surfaces under severe loads, damaged articular cartilage can hardly self-regenerate, which results in prolonged pain and functional restrictions on the joints [2]. The most common strategies in clinical treatments related to cartilage defects include microfracture surgery, osteochondral autografts, matrix-induced autologous chondrocyte implantation, and autologous chondrocyte implantation [3]. However, in the above techniques, incomplete cartilage development can lead to persistent joint damage or poor prognosis. Other constraints, such as a lack of available chondrocytes and low efficacy, are frequently noted in older patients [4]. As the number of patients with cartilage defects increases, the concern over public health and the health system has become more pressing. Given limited treatment techniques, it is urgent to explore alternative treatment strategies.

Hydrogels have gained a great deal of interest in cartilage tissue engineering to further expand the use of biomaterials in cartilage repair. A hydrogel is a three-dimensional hydrophilic polymer network that is capable of absorbing a large amount of water without losing its structure [5]. For the production of hydrogels, there is a vast selection of available source materials. The use of synthetic polymers in tissue engineering has been extensively researched. While synthetic polymer hydrogels can perform well mechanically and have good reproducibility, they can be problematic because of their often underestimated biocompatibility concerns [6]. In recent years, hydrogels derived from natural sources have been considered a potential candidate for cartilage tissue engineering because of their preferred biocompatibility, safety, and stability. The advantages of hydrogels can promote the growth, proliferation, and differentiation of chondrocytes and support cartilage regeneration [7].

Natural materials derived from animal cartilage tissues, such as acellular cartilaginous matrix (ACM), are gaining increasing attention for the treatment of cartilage defects because they provide an ideal extracellular matrix (ECM) [8]. ECM in native cartilage tissues crucially regulates chondrocyte behaviors and maintains tissue functions. The cartilage ECM is predominantly made of collagens and other critical molecules in the microenvironmental niche, such as glycosaminoglycans, proteoglycans, and growth factors [9]. However, how to retain these molecules in the hydrogel is yet to be elucidated in the engineering of cartilage tissue.

Several recent studies reported that the ACM of decellularized tissues could be solubilized in pepsin and subsequently polymerized into hydrogels under physiological conditions. Sun et al. [10] discovered that ACM hydrogels remained a part of the biologically active molecules found in native tissues and showed significant therapeutic potential in remodeling source tissues after implantation. A study by Hsieh et al. [11] showed acidic treatments, such as HCL, could solubilize the cellular cytoplasmic components, remove nucleic acids (for instance, DNA), preserve the structure and function of the native ECM, and simultaneously disinfect the material through entering microorganisms and oxidizing microbial enzymes. However, ACM-based hydrogel has a short biodegradation period and low physic-mechanical properties, so its application is rather limited in practical situations. Different concentrations of ACM and HCl, as well as the size of ACM, can influence the hydrogel’s characteristics. In order to solve certain limitations, this study has made an effort to develop ACM-based hydrogels from porcine cartilage tissue, which may enable a one-time operation, reduce the risk of surgeries, and improve medical efficiency. This study can be divided into two major parts: firstly, we designed and characterized ACM hydrogels. Next, we evaluated the tissue-adhesive properties of the prepared ACM hydrogels in an ex vivo test.

## 2. Materials and Methods

### 2.1. Chemical Reagents and Drugs

Pepsin (Sigma-Aldrich, St. Louis, MO, USA), Hoechse33258 (Sigma-Aldrich, MA, USA), Papain (Sigma-Aldrich, MO, USA), HCl (SHOWA, Gyoda, Japan), NaOH (SHOWA, Japan), Triton X-100 (Sigma-Aldrich, MO, USA), Aprotinin (Sigma-Aldrich, MO, USA), RNase A (Sigma-Aldrich, MO, USA), Chloramine T trihydrate (Sigma-Aldrich, MO, USA), DNase I (Merck, Darmstadt, Germany), Citric acid (JTbaker, Radnor, PA, USA), Sodium acetate trihydrate (JTbaker, PA, USA), and Quant-iTTM PicoGreen dsDNA Reagent and Kits (Invitrogen, Waltham, MA, USA) were used.

### 2.2. Preparation of ACM Powder and Decellularization

The ACM powder was prepared using the method described by Park et al. [12], with modifications. Fresh cartilage tissues were harvested from pigs sacrificed in the early morning on testing day from a local traditional market. The knee joint cavity was incised with a scalpel and tissue scissors to expose the joint tibia and femur and to remove excess connective tissue. Then, the cruciate ligament was excised with a scalpel to expose the cartilage surface and sprayed with DPBS to keep it moist. The cartilage was removed using a scalpel, with length and width controlled to be less than 0.5 cm, and the thickness was less than 1 mm. The cartilage was then placed in a centrifuge tube with DPBS containing 1% penicillin-streptomycin-neomycin (PSN) antibiotic mixture (15640055, Thermo Fisher Scientific, Waltham, MA, USA). The cartilage fragments were washed twice with DPBS and centrifuged at 1200 rpm for 10 min to remove the supernatant. After adding 10 mL of double-distilled water (ddH_2_O) to each tube, they were placed in a −80 °C refrigerator for 5 h and then freeze-dried for 48 h (FD21-3S-12P, KINGMICH, New Taipei City, Taiwan). After freeze-drying cartilage fragments, 0.5 g fragments were poured into sieves with different meshes, shook for 10 min on a sieve shaker (VIB-RO, SHIN KWANG, New Taipei City, Taiwan), and separated into large (L) (0.25~0.50 mm), medium (M) (0.10~0.25 mm), and small (S) groups (0.05~0.10 mm).

The decellularization of the ACM tissue was achieved by adding 0.35 g of decellularized cartilage matrix material to 10 mL of ddH_2_O and then freezing it at −80 °C for 5 h. Afterwards, 50 mL of decellular solution, containing aprotinin (10 KIU/mL), Triton X-100 (1%, (*v*/*v*)), DNase (50 units/mL), and Rnase (1 Kunitz units/mL), was added to each tube. Decellularization was performed by sonicating at 37 °C for 12 h. Following the removal of the cells, the ACM was centrifuged at 2000 rpm for 10 min, shook, and washed with ddH_2_O for 15 min until no foam was visible. After extracting the supernatant, 10 mL of ddH_2_O was added, and the tubes were frozen at −80 °C for 5 h, freeze-dried for 48 h, exposed to UV irradiation for 24 h, and placed in a drying oven.

### 2.3. Preparation of ACM Hydrogels

In this study, ACM hydrogels were prepared mainly via the enzymatic digestion method described by Bhattacharjee et al. [13]. The ACM was digested in a porcine pepsin solution in a ratio of 10:1. Pepsin was used at a concentration of 1 mg/mL using 0.01 N HCl, 0.1 N HCl, and 1 N HCl. The mixture was then stirred at room temperature for 48 h. The ACM digests were neutralized to pH of 7.4 by adding 0.1 N, 1N, and 10 N NaOH, followed by 10 × PBS. The neutralized ACM was then diluted to the desired final ACM concentration (10, 20, and 30 mg/mL), with 10 × PBS on ice. The ACM pre-hydrogels were then placed in an incubator heated to 37 °C to form hydrogels.

### 2.4. Histological Analyses of Decellularized ACM

Decellularized ACM of different sizes was fixed on glass slides, stained with Hoechst33258 dye at a concentration of 1 g/mL for 30 min in the dark, and then rinsed twice with ddH_2_O. Samples were dried at room temperature in the absence of light. Nuclei were examined using fluorescence microscopy (Eclipse 50i, Nikon, Tokyo, Japan), with a wavelength of UV absorption λ_max_ = 343 nm.

### 2.5. Biochemical Quantification

Various biochemical quantification methods were used to determine the biochemical composition of decellularized ACM and ACM hydrogels, including DNA, glycosaminoglycan (GAG), and collagen contents. Prior to analysis, the tissues were lyophilized, and 1 mg of dry ACM tissue was digested at 60 °C with 3 units/mL papain. The measured values were normalized to the dry weight of the samples.

#### 2.5.1. DNA Quantification

Following the manufacturer’s instructions, DNA was isolated and quantified using a Quant-iT PicoGreen dsDNA assay kit. Briefly after papain digestion, DNA samples were mixed with the Quant-iT PicoGreen reagent, and the samples were excited at 485 nm, and the fluorescence emission intensity was measured at 530 nm with a spectrofluorometer (Thermo Scientific Varioskan Flash, Thermo Fisher Scientific, Waltham, MA, USA).

#### 2.5.2. GAG Quantification

To directly measure the sulfated GAG content, papain digested samples were stained with 1,9-dimethylmethylene blue (DMB) and photometrically (Varioskan Flash-5250040, Thermo Fisher Scientific, Waltham, MA, USA) measured at 525nm, as described. A dilution series of chondroitin sulfate in PBS was used as the standard solution [14].

#### 2.5.3. Collagen Quantification

The samples were dried and analyzed using a hydroxyproline assay kit and using hydroxyproline as a standard [15,16].

#### 2.5.4. Solubilization of Decellularized ACM and ACM Hydrogels

With 20 mg of pepsin diluted in 10 mL of 1 N HCl, 200 mg of decellularized ACM in the three distinct sizes were mixed. After stirring them for 48 h at 4 °C, the digestion of the decellularized ACM was observed with the naked eye. The entirely digested decellularized ACM was screened out, as it has been fully digested under the action of a fixed acidic environment and enzymes. In the next step, we evaluated the function of three different amounts of ACM (100, 200, and 300 mg). They were digested with three different pepsin concentrations (10, 20, and 30 mg) in HCL solutions (0.01 N, 0.1 N, and 1 N), respectively. Hydrogels were evaluated using test tube inversion in a 15-mL test tube at 4, 25, and 37 °C, following the same procedure as previously reported [17].

### 2.6. Turbidimetric Gelation Kinetics

The gelation kinetics of ACM hydrogels were evaluated turbidimetrically, as described by Gong et al. [18]. Briefly, the ACM hydrogels were plated in 96-well plates (100 μL/well) at 4 °C in concentrations of 4, 8, and 10 mg/mL. In a spectrophotometer (UV-260, Thermo Fisher Scientific, Waltham, MA, USA), absorbance was measured every 2 min for 1 h at 405 nm. With Equation (1), the normalized absorbance (NA) was calculated using the scaled measurements from 0% (at time 0) to 100% (at the maximum absorbance). *P* is the absorbance at a given time, *P*_0_ is the initial absorbance, and *P_max_* is the maximum absorbance. The time required to reach 50% of *P_max_* is defined as *t*_1/2_ and the gelation rate (S) represents the slope of the linear region of the gelation curve.
(1)$$\mathrm{NA}=\frac{P-P_{0}}{P_{max}-P_{0}}$$

### 2.7. Rheological Studies of ACM Hydrogels

The rheological measurements of the ACM hydrogels (4, and 10 mg/mL) were carried out by a rheometer (Physica MCR 301, Anton Paar, Graz, Austria) using the oscillatory shear stress method, following the same procedure as previously reported [19]. A parallel plate was used with a plate diameter of 50 mm and a plate gap of 102 μm. The storage modulus (G′) was calculated by controlling the frequency (0.159 Hz), stress-strain (1 Pa), and the temperature (37 °C), as well as by increasing the time duration from 0 to 30 min.

### 2.8. Ex-Vivo Experiment

The ex vivo experiment was performed according to previously developed method [20]. Using bone extraction instruments, a 10 mm diameter and 0.1 cm depth defect were created above cartilage tissues from pigs purchased from a local traditional market, which were then treated with 20 mg/mL and 30 mg/mL of ACM hydrogels, respectively. After hydrogels were injected, the experiments were randomized into six groups (N = 10 in each group) to evaluate the stability of hydrogel within bone tissue: (A) rest at 37 °C, 20 min; (B) resuspend at room temperature, 20 min; (C) washed with DPBS, 5 times; (D) dipped in water, 20 min; (E) shook with100 rpm for 20 min; (F) sonicated for 20 min.

### 2.9. Statistical Analysis

Data were expressed as the mean ± standard deviation. Statistical significance was assessed by a two-way analysis of variance (ANOVA), followed by Tukey’s test by Microsoft Excel. Statistical significance was considered at three levels: * *p* < 0.05, ** *p* < 0.01, and *** *p* < 0.001.

## 3. Results

### 3.1. Preparation and Characterization of Decellularized ACM

Figure 1 displays the schematics of the ACM preparation process. ACM was effectively prepared after a series of decellularization procedures, and the effect of decellularization was further investigated. As demonstrated by the red arrows, the Hoechst33258 staining of decellularized ACM fragments of various sizes (Figure 2) revealed that the cartilage fragments include numerous bright spots, which are cells. In contrast, the ACM groups successfully eliminated a considerable number of the bright spots, but they still maintained the natural chondrocyte cavities (lacunae), as marked by the yellow arrows. Following the cell removal method, all three sizes of ACM eliminated the majority of cells.

As shown in Figure 3a, DNA quantitative analysis further confirmed that more than 98% of DNA content was removed, and the remaining DNA concentration is lower than 50 ng/mg after decellularization. To confirm ECM preservation, the contents of GAG and collagen components in ACM were quantified, with the results showing that 78.28% of the GAG components were retained in the L group, 76.08% were retained in the M group, and 63.80% were retained in the S group, compared to the control group (samples prepared without decellularized procedure, 100%) (Figure 3b). Thus, the results suggested that during decellularization, GAG is more readily lost when the particle size of the decellularized matrix is small.

Furthermore, collagen is the main component of articular cartilage, and it can be determined with hydroxyproline. When compared with normal cartilage fragments (control group), ACM after decellularization could retain at least 80% collagen compared to control (samples prepared without decellularized procedure, 100%), as shown in Figure 3c. Size S groups showed no statistically significant amount of collagen content compared to the control group. Moreover, the solubility tests of different sizes of decellularized ACM powder in pepsin revealed that the S group of decellularized ACM had no powder, no precipitation, and was uniformly dispersed in the solution after contacting with the enzyme solution due to its larger unit surface area (Figure 4). In contrast, the L and M groups of ACM did not completely disperse or dissolve. Based on the reasons above, the S group of decellularized ACM was selected in our study for further preparation of the ACM hydrogels.

### 3.2. Preparation and Characterization of ACM Hydrogels

In this study, the digest effect of enzymes in two groups of ACM—the fragment group (control) and the ball-milled group—was screened first at three different temperatures, namely, 4, 25, and 37 °C. The result showed that, at 4 °C, the ball milling group could control the particle size of the matrix, and the decellularization effect was better than that of the fragment group. However, at 25 and 37 °C, both groups were not agglomerated. According to the findings, improved gelation could occur when the digestive reaction parameters for magnet stirring were 320 rpm, 200 mg of the matrix, and 20 mg of the enzyme. The results also demonstrated, in Figure 5a, that the gelation effect of the decellularized ACM was enhanced when the enzymatic reaction was conducted in the presence of a higher concentration of HCl (1 N > 0.1 N > 0.01 N). Similarly, the higher the concentration of NaOH, the more hydrogel that could agglomerate. At the concentration of 10N, the consequent was at its best (Figure 5b).

The findings further suggested that increasing ACM hydrogel concentration improved gelation. The optimal gel state was reached when the concentration of ACM hydrogels was 30 mg/mL. In the gelation process, the gel could be formed by pre-gelling at room temperature (about 25 °C) for 3 min. The data have confirmed that the pH range of the colloid should be between pH 6.8 and 7.6 in the pre-gelling stage, with pH 7.4 being optimal, and 1/9 of 10 X PBS was the most appropriate salt addition.

The amount of DNA, collagen, and GAG contents of ACM hydrogels were measured and then compared with samples prepared without decellularized procedure (control group) and the S group of ACMs. Figure 6 indicated that 99.61% DNA in ACM hydrogels was significantly removed (*p* < 0.001, Figure 6a) and still retained 61.18% of GAG (compared with control samples set as 100%, *p* < 0.001, Figure 6b) and 78.29% of collagen content (compared with control samples set as 100%, *p* < 0.01, Figure 6c). All ACM hydrogels demonstrated significant differences at the 0.05% level.

The gelation kinetics of ACM hydrogels at 4, 8, and 10 mg/mL were evaluated using a normalized absorbance to define the lag phase, the time to reach 50% of the final turbidity, and the time to reach complete gelation. The turbidimetric gelation kinetics for all concentrations showed sigmoidal shapes (Figure 7a). The differences observed in the kinetic curve shapes indicated a shorter lag phase for ACM hydrogels at 10 mg/mL than the gels at 8 mg/mL or 4 mg/mL. The time to reach 50% gelation was also shorter for ACM hydrogels at 10 mg/mL (T1/2 = 10 min) compared with the lower concentrations (T1/2 (8 mg/mL): 20 min; T1/2 (4 mg/mL): 25 min). It also took ACM hydrogels at 10 mg/mL significantly less time (T1/2 = 10 min) than those with lower concentration, i.e., T1/2 (8 mg/mL) is 20 min and T1/2 (4 mg/mL) is 25 min.

Parallel plate rheometers were used to determine the rheological characteristics of ACM hydrogels. Figure 7b displayed the frequency sweep tests of storage modulus (G′) within a range of 0–30 min at a fixed frequency (0.159 Hz), stress-strain (1 Pa), and temperature (37 °C). The G′ values of ACM hydrogels increased after ACM pepsin digests were neutralized. In addition, the rheological properties of hydrogels were dependent on the content of ACM. The elastic modulus remained constant as time progressed, and the modulus between 10 mg/mL and 4 mg/mL was relatively strong. As the rigidity of a hydrogel system network was defined by its G′ values, these results were attributed to the stabler and denser internal structure of the ACM hydrogels, demonstrating that the higher the colloid concentration, the better the mechanical condition.

### 3.3. Tissue-Adhesive Property of the ACM Hydrogels (Ex-Vivo Test)

To further scrutinize the integration of ACM hydrogels and bone tissue under different conditions, the adhesion strength of the ACM hydrogels for 20 mg/mL and 30 mg/mL concentrations were evaluated by measuring the stability of hydrogels within bone tissue (Figure 8 and Table 1). The tissue-adhesion test with ACM hydrogels revealed that 30 mg/mL concentration could adhere better to the tissue surface than the 20 mg/mL concentration. Although the 20 mg/mL hydrogels partially flaked off the surface after being washed with DPBS, there was still gel residue. In contrast, the visual appearance showed that a concentration of 30 mg/mL maintained excellent adherence to the tissue at all times. This revealed that, when the hydrogel concentration was increased, the hydrogel could attach to the tissue more stably.

## 4. Discussion

Up to now, even though hundreds of cartilage repair materials have been continually developed by scientists, the most suitable biological scaffold for cell growth has not yet been found. Cartilage is a connective tissue composed of chondrocytes and ECM. In cartilage tissue, all nutrients and metabolites must be exchanged by diffusion and transferred to chondrocytes through the ECM. The ECM material is an ideal tissue scaffold, and its clinical application is limited by autologous therapy, resulting in insufficient sources [21]. However, cells or antigens present in xenogeneic or allogeneic materials may trigger adverse immune responses. Therefore, decellularization aims to remove xenogeneic nucleic acids, antigens, etc., to avoid adverse severe immune responses [22]. The main factors inducing severe immunity are cell surface antigens and nucleic acid substances. Researchers have used physical, chemical, and enzyme methods to remove cell-related substances and to retain ECM components and structures to prepare decellularized matrix materials [23].

Compared to other tissues, the structure of cartilage tissue is denser, making it more difficult to remove cellular immunogens and retain natural matrix components [24]. Kridel et al. [25] used artificial dermis to repair the perforation of nasal septal cartilage in patients. After three months of observation, it was found that the repairment was similar to that of autologous tissue. Porzionato et al. [26] employed alcohol dehydration, freeze-drying, and repeated freezing and thawing to decellularize rabbit nasal septal cartilage and then implanted chondrocytes as scaffolds. However, because cartilage tissue still retained considerable antigenicity, implanting it into living bodies was prone to adverse immune responses. Thus, Narez et al. [27] used non-ionic surfactants to treat meniscal cartilage tissue. Although it could maintain structural integrity, it could not entirely remove the cells.

Chemical and enzymatic methods were used to prepare decellularization solution, combined with physical methods, frozen cartilage powder, and ultrasonic shock to remove cells. Regarding the surface antigen, it has been pointed out that violent hyperacute rejection is elicited when native or incompletely decellularized porcine heart valves are transplanted into humans, similar to the reactions in ABO incompatibility mainly because of the surface residue alpha-gal epitope (galactose-alpha1-3-galactose) on the porcine cell [28]. However, a few studies have pointed out that allograft cartilage does not induce apparent immune rejection [29,30]. Moreover, Revell and Athanasiou [31] showed that a large amount of cartilage matrix could reduce the contact area between the host T-cell and the antigen of the source chondrocyte so that the immune response is reduced or does not occur. Therefore, the success rate of tissue repair can be improved. The in vivo findings of Huey et al. [32] pointed out that, as long as the alpha-gal epitope can be successfully removed in the cartilage transplantation experiment, no other immune rejection will occur.

The threshold concentration of residual cellular material within ECM sufficient to elicit adverse host responses has been suggested based on some in vivo studies that show that adverse cell and host immune reactions have been avoided. For example, dsDNA per mg ECM dry weigh should be less than 50 ng, or there should be lack of visible nuclear material in tissue sections [33]. The ACM and ACM hydrogel produced in this study fit the criteria that, after the cell removal procedure, most cells and DNA were removed from the cartilage matrices of all three sizes. (Figure 2, Figure 3a and Figure 6a).

By comparing the decellularized ACM of different sizes obtained by ball milling with the normal cartilage fragments of the control group, this study successfully retained at least 80% of the collagen after the decellularization process in each group of ACMs. The collagen content of the S group, with the smallest particle size, was not statistically different from the cartilage fragment group, and it also retained the most collagen; therefore, the S group of decellularized ACMs was the most suitable for further ACM hydrogel preparation. Compared with decellularization using sodium dodecyl sulfate, Kheir et al. [34] also removed more than 98% of DNA, but retained less than 2% of GAG. The decellularization method used in the current study combined physical, chemical, and enzymatic methods, with the minimum GAG retention being 63.80% and the maximum being 78.28%.

Furthermore, this study evaluated the dissolution of different-sized ball-milled ACM and sieved decellularized ACM following pepsin digestion in HCl. Pepsin digestion of different ACM sizes also showed different results, which ultimately influenced the gelation status. When the pepsin fully digests the collagen to remove both telopeptides, it can be dissolved in an acidic solution [35]. The digestion result was observed precisely 48 h after digestion. The S group of ACMs had no powder, no precipitation, and was uniformly dispersed in the solution because of its larger unit surface area. It is worth noting that the sterilization of the biomaterials would be a critical issue that could change the biological or mechanical properties of the materials [36]. There have been many techniques proposed to disinfect biomaterials, including peracetic acid and/or ethanol, antimicrobials, UV, ethylene oxide, gamma radiation, etc. [37]. In this study, ACM was disinfected by UV exposure and mixed with sterilized reagents to obtain ACMH for further examinations.

During the preparation of ACM hydrogels for tissue, the colloid characteristics might be affected by any parameter [38]. In this study, the ACM of the ball-milled group was used to study the digest effect of enzymes at different temperatures. The findings demonstrated that the matrix size could be regulated at 25 °C, and the decellularization effect was superior to that of the fragment group. Results from digestion demonstrated that an acidic environment could influence gel performance positively. Improved gelation was also observed when the matrix was stirred, when the matrix concentration was raised, and when enzymes were used for digestion.

Pepsin, depending on the pH of its surrounding environment, can engage in a variety of actions, and its capacity to digest ACM can also change [39]. Although enzymatic digestion was more complete in 1 N HCl than in the acid, experimental data showed that there were still particles in 0.01 N HCl and 0.1 N HCl that have not been entirely digested. As a result of complete digestion, the 1 N HCl group exhibited higher self-cohesion throughout the gelation phase. Self-reorganization was observable, and the gelation state of the gel improved. The results showed that the gelation effect of decellularized ACM was better under the action of enzymes with a higher concentration of HCl. Besides, the colloid could form a gel within a specific pH range in the pre-gel stage. The approximate range was between pH 6.8 and pH 7.6, with pH 7.4 being the optimal value. Furthermore, 1/9 volume of 10 X PBS was the most appropriate salt addition, which could effectively help the self-reorganization phenomenon of the hydrogel.

Rheology was used to study the viscoelastic properties of ACM hydrogels. It is interesting to note that, regardless of the tissue origin, hydrogels exhibited similar sigmoidal-shaped gelation profiles. Although the specific components responsible for the gelation of these ACM hydrogels were unclear, the high soluble collagen content of the material suggested that gelation was most likely due to the presence of collagen molecules [40]. The collagen monomers could aggregate and self-assemble into thin filaments, which could then crosslink into collagen fibers that interweave with themselves and other ECM components to contribute to hydrogel formation [41]. The turbidity data measured here showed that, during hydrogel assembly, the 10 mg/mL CM hydrogels reached a steady state plateau faster than the 8 or 4 mg/mL ACM hydrogels. It can be observed that the gelation time decreases with increasing colloid concentration, which is similar to that of pure type I collagen in the tissue gel [42]. The rheological characterization of ACM hydrogels obtained in this study had similar properties to tissue glue prepared from muscle matrix and was significantly different from the tissue glue of the meniscus, dermis, and bladder matrices [43].

In terms of the elastic modulus, the dermal matrix gel is approximately 466 Pa at 8 mg/mL [44], the bladder matrix is roughly 182 Pa [44], and the muscle matrix is around 6.5 Pa [45]. In this study, the G′ values of 10 mg/mL of ACM hydrogel were about 20 Pa, and the mechanical properties were lower than the dermal matrix gel, as well as bladder matrix. It was evident from the data that, as time elapsed, the elastic modulus between 10 mg/mL and 4 mg/mL was relatively stronger than others; thus, we could conclude that the higher the colloidal concentration, the better the mechanical condition.

The biggest challenge in cartilage tissue engineering is the poor integration of the implanted scaffold into the native tissue arising from the lack of tissue adhesion [46]. In this study, a higher concentration of hydrogel maintained excellent adherence to the tissue. Our ACM hydrogels can firmly attach to peptides and proteins on the tissue surface and can form covalent bonds with such nucleophilic functional groups of proteins present on the tissue surfaces. In addition, noncovalent interactions, such as hydrogen bonds, hydrophobic interactions (π–π interaction), cation–π interactions, and electrostatic interactions could occur between ACM hydrogels and tissue surfaces [47,48].

The acellular matrix can systematically repair damaged wounds or regenerate tissue with a similar structure to the original [49]. There are more and more biologically derived decellularized products on the market; however, there are no products related to decellularized cartilage matrix. Therefore, this study combined physical, chemical, and enzymatic methods and attempted to achieve the potential final product. To investigate the effects of different sizes of cartilage powder on forming hydrogel, we prepared size S, M, and L particles by ball milling machine for testing. The yield of S size powder was about 30% of the total cartilage powder by the parameter we set for the machine (data not shown). The results further showed that the S group is the most suitable size for further study. For future plans, although the yield of S size particle in this study was only about 30% from the original material, we can increase the yield of S size powder by tuning the setting of the ball milling machine for later mass production procedures. The optimized setting of the parameter needs to be further examined. Furthermore, the structure of the hydrogels will be examined by scanning electron microscope to further investigate the details of the hydrogel. The cytotoxicity and immunogenicity of the cartilage-derived hydrogel will also be evaluated to address the safety issue of this material by testing with human primary chondrocytes. The future works will focus on the cartilage repair effects by testing ACM hydrogel with human primary chondrocytes and cartilage defect animal models.

## 5. Conclusions

In this study, parameters for preparing ACM hydrogel were investigated and discussed. Despite decellularization, 61.18% of GAGs and 78.29% of collagen remained in the ACM hydrogels compared to the samples without decellularization (100%). The decellularized ACM was used to produce the hydrogel, and turbidity gel kinetics confirmed that the increased hydrogel concentration contributed to a shorter gelation time. In the rheological investigation, ACM hydrogel had lower mechanical properties than other tissue glues; however, the colloidal mechanical characteristics positively correlated with the concentration of hydrogels. A series of physical tests showed that ACM hydrogel had an excellent adhesion effect on cartilage tissue. We strongly believe that this study offers useful information for developing and manufacturing ACM hydrogel to serve as a potential alternative scaffold for future cartilage defect treatment.

## Figures and Tables

**Figure 1 jfb-13-00279-f001:**
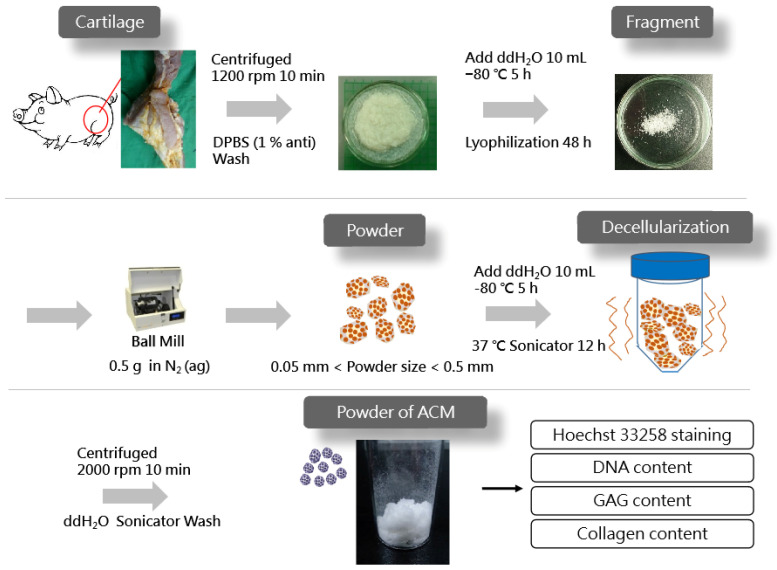
Preparation of acellular cartilage matrix (ACM) powder.

**Figure 2 jfb-13-00279-f002:**
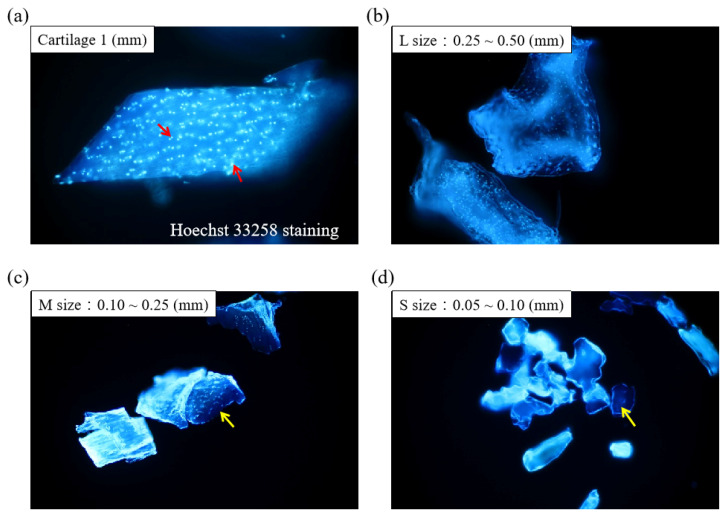
Histological analyses of different sizes of decellularized ACM using Hoechst 33258 staining (200×). Red arrow: cell nuclei; yellow arrow: lacunae.

**Figure 3 jfb-13-00279-f003:**
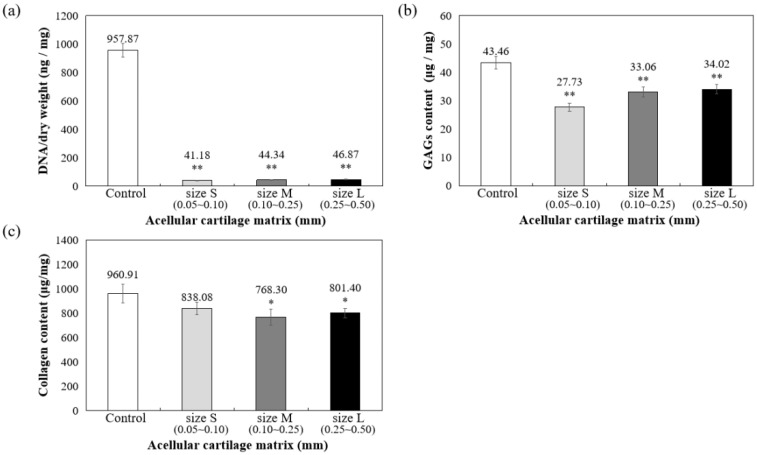
Quantification of (**a**) residual DNA, (**b**) glycosaminoglycans (GAGs), and (**c**) collagen content in decellularized ACM powder. * *p* < 0.05, and ** *p* < 0.01 when compared to control (samples prepared without decellularized procedure).

**Figure 4 jfb-13-00279-f004:**
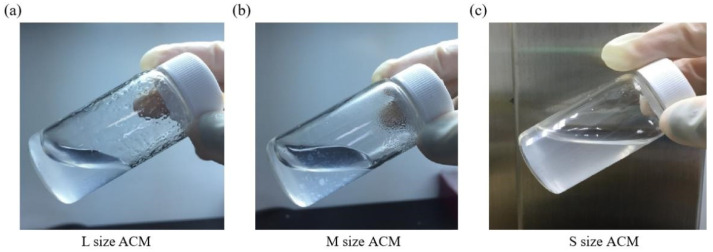
Pepsin digestion test for (**a**) L size, (**b**) M size, or (**c**) S size of ACM powder.

**Figure 5 jfb-13-00279-f005:**
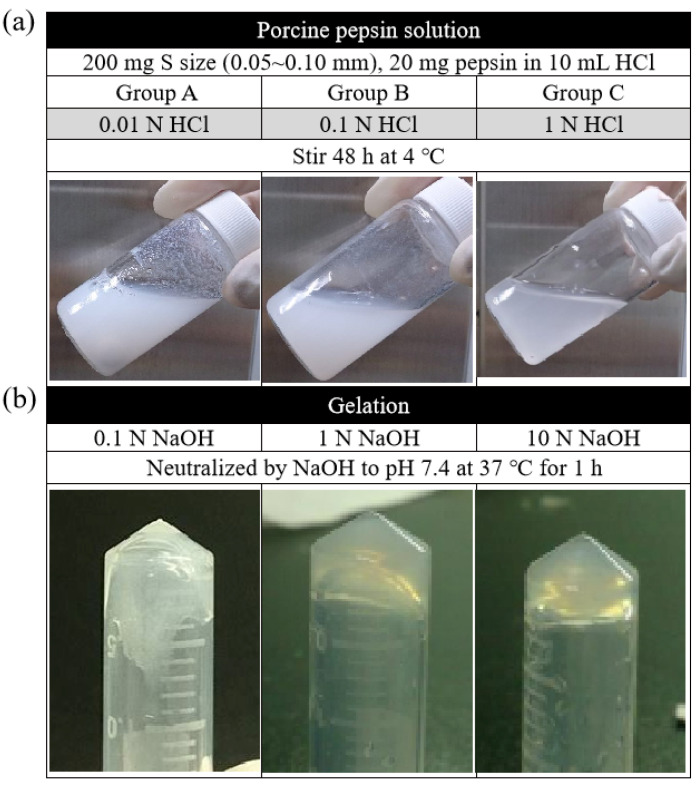
(**a**) Digestion of ACM in different concentrations of HCl and (**b**) neutralization of ACM digests in pH of 7.4 by adding different concentrations of NaOH.

**Figure 6 jfb-13-00279-f006:**
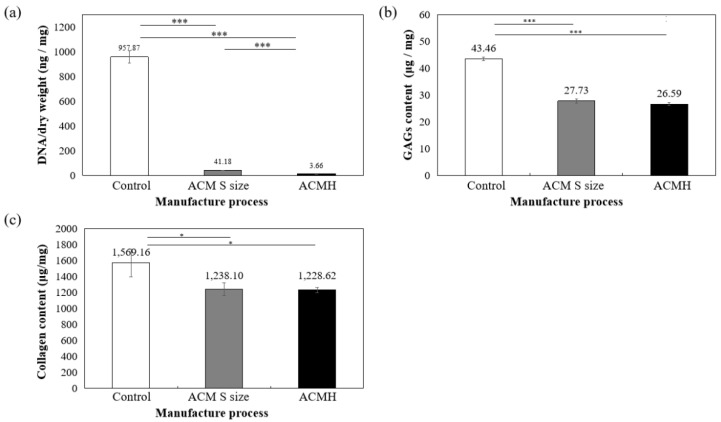
Quantification of (**a**) residual DNA, (**b**) glycosaminoglycans (GAGs), and (**c**) collagen content in ACM hydrogel. * *p* < 0.05 and *** *p* < 0.001 when compared between groups. Control: samples prepared without decellularized procedure.

**Figure 7 jfb-13-00279-f007:**
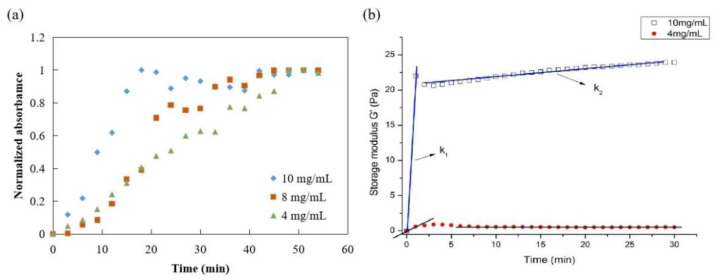
Mechanical properties of ACM hydrogel: (**a**) turbidimetric gelation kinetics of ACM hydrogels, (**b**) rheological properties of ACM hydrogel (storage modulus).

**Figure 8 jfb-13-00279-f008:**
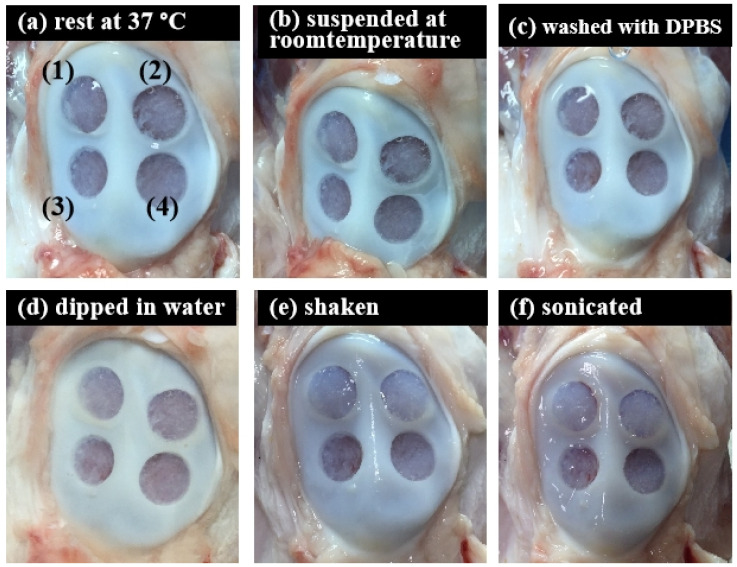
Tissue-adhesive test of the ACM hydrogels (ex vivo test) in different conditions: (**a**) rest at 37 °C, 20 min; (**b**) suspended at room temperature, 20 min; (**c**) washed with DPBS, 5 times; (**d**) dipped in water, 20 min; (**e**) shook with 100 rpm for 20 min; and (**f**) sonicated for 20 min.

**Table 1 jfb-13-00279-t001:** Physical test chart of ACM hydrogels and cartilage patella defect.

No.	ACMH Concentration (200 μL)	(a) Rest at 37 °C	(b) Suspended at Room Temperature	(c) Washed with DPBS	(d) Dipped in Water	(e) Shaken	(f) Sonicated
(1)	30 mg/mL	o	o	o	o	o	o
(2)	30 mg/mL	o	o	o	o	o	o
(3)	20 mg/mL	o	o	x	x	x	x
(4)	20 mg/mL	o	o	o	o	o	o

## Data Availability

The data presented in this study are available on request from the corresponding author.

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
