# Peer review of "Preparation and Characterization of Extracellular Matrix Hydrogels Derived from Acellular Cartilage Tissue"

_jfb, 2022, doi:10.3390/jfb13040279_

Round 1
Reviewer 1 Report
Overall suggestion: Accept with minor changes
Summary: The introduction is well-written and the aims of the paper is clearly outlined. The methods are described in detail and the results have been presented well. Discussion is sound and will be useful to add a separate paragraph on ‘limitation’ acknowledging the limitation the studies.
Minor revisions:
1. Please add the full form of ‘dd’ in ddH2O in line 105, even though it seems clear to most readers
2. In section 2.9, could you please indicate what softwares you have used to generate your graphs for your data? Graph pad Prism? Excel? Origin?
3. Please acknowledge the use of male pigs only in your study outlining the absence of female pigs as a potential limitation to the study in a separate paragraph in the discussion.

Author Response
Response to Reviewer 1 Comments
Dear editor and reviewer,
Thank you for the review of the manuscript entitled "Preparation and characterization of extracellular matrix hydrogels derived from acellular cartilage tissue" (jfb-2053759). We thank the constructive comments and have revised our manuscript accordingly in the text. The responses are highlighted in red font and listed below. Thank you for the prompt attention.
Yours sincerely,
Professor Hsu-Wei Fang, Ph.D. (Corresponding author)
Department of Chemical Engineering and Biotechnology, National Taipei University of Technology. Taipei 10608, Taiwan. (hwfang@ntut.edu.tw)
Reviewer 1 Comments
Summary: The introduction is well-written and the aims of the paper is clearly outlined. The methods are described in detail and the results have been presented well. Discussion is sound and will be useful to add a separate paragraph on ‘limitation’ acknowledging the limitation the studies.
Minor revisions:
Point 1: Please add the full form of ‘dd’ in ddH2O in line 105, even though it seems clear to most readers
Response 1:
Thanks for the reminder. We have added double-distilled water (ddH2O) in the manuscript, please check section 2.2, line 108: “After adding 10 mL of double-distilled water (ddH2O) to each tube, they were placed in a -80 °C refrigerator for 5 hours and then freeze-dried for 48 hours (FD21-3S-12P, KINGMICH, New Taipei City, Taiwan).”
Point 2: In section 2.9, could you please indicate what softwares you have used to generate your graphs for your data? Graph pad Prism? Excel? Origin?
Response 2:
We used Microsoft Excel for graphs and statistics and we have added this information in the manuscript, please see section 2.9, line 203.
Point 3: Please acknowledge the use of male pigs only in your study outlining the absence of female pigs as a potential limitation to the study in a separate paragraph in the discussion.
Response 3:
Thanks for the reminder. We corrected the description of the method that the ex-vivo experiment was conducted with tissues purchased from local market and we actually did not know which gender they were. For the future plan with in vivo animal tests, we will use both male and female animals to evaluate the effects of ACM hydrogels. Please check section 2.8 line 193: “The Ex-vivo experiment was performed according to previously developed method [20]. Using bone extraction instruments, a 10 mm diameter and 0.1 cm depth defect was created above cartilage tissues from pigs purchased from local traditional market”

Reviewer 2 Report
In this research, the authors have prepared hydrogel from decellularized cartilage tissue. The optimum procedure has been suggested and the hydrogel was characterized. Overall, these findings of the study were quite useful and interesting. The manuscript was well-written and coherent structure. However, I recommend some points to improve quality of the manuscript.
1. The yield of the procedure for preparing S size ACM should be provided and discussion in degree of efficiency with other relating works.
2. The SEM measurement for hydrogel need to be conducted.
3. In the rheological measurement, the author should include the loss modulus in parallel with storage modulus, and the gelation point need to determine.
4. If possible, essential experiments as cytotoxicity and cytocompatibility need to be complement.
Author Response
Response to Reviewer 2 Comments
Dear editor and reviewer,
Thank you for the review of the manuscript entitled "Preparation and characterization of extracellular matrix hydrogels derived from acellular cartilage tissue" (jfb-2053759). We thank the constructive comments and have revised our manuscript accordingly in the text. The responses are highlighted in red font and listed below. Thank you for the prompt attention.
Yours sincerely,
Professor Hsu-Wei Fang, Ph.D. (Corresponding author)
Department of Chemical Engineering and Biotechnology, National Taipei University of Technology. Taipei 10608, Taiwan. (hwfang@ntut.edu.tw)
Reviewer 2 Comments
In this research, the authors have prepared hydrogel from decellularized cartilage tissue. The optimum procedure has been suggested and the hydrogel was characterized. Overall, these findings of the study were quite useful and interesting. The manuscript was well-written and coherent structure. However, I recommend some points to improve quality of the manuscript.
Point 1: The yield of the procedure for preparing S size ACM should be provided and discussion in degree of efficiency with other relating works.
Response 1:
Thanks for the comment and it is a great point that we need to know for further development of the hydrogel. We added this paragraph in section 4 for discussion:
In this study, to investigate the effects of different sizes of cartilage powder on forming hydrogel, we prepared size S, M, and L particles by ball milling machine for testing. The yield of S size powder was about 30 % of the total cartilage powder by the parameter we set for the machine (data not shown). The results further showed that S group is the most suitable size for further study. For future plan, although the yield of S size particle in this study was only about 30 % from original material, we can increase the yield of S size powder by tuning the setting of the ball milling machine for later mass production procedure. The optimized condition of the machine needs to be further examined. (section 4, line 445-453)
Point 2: The SEM measurement for hydrogel need to be conducted.
Response 2:
Thanks for the comment. Indeed the morphology and structure of the hydrogel need to be addressed. We proposed to conduct SEM test in the discussion section for our further plan. Please check section 4, line 453-455.
Point 3: In the rheological measurement, the author should include the loss modulus in parallel with storage modulus, and the gelation point need to determine.
Response 3:
Rheology is typically utilized to determine the storage modulus, or stiffness, of the hydrogel following gelation. By rheological measurement, the loss modulus was relatively low as base line in our study. From an article regarding hydrogel derived from decellularized dermal extracellular matrix, similar result was also observed and only storage modulus was shown (Wolf et al., 2012). The half gelation time of 10 mg/ml ACMH was about 10 min, 8 mg/ml ACMH was 20 min and 4 mg/ml ACMH was 25 min.
Reference:
Wolf MT, Daly KA, Brennan-Pierce EP, Johnson SA, Carruthers CA, D'Amore A, Nagarkar SP, Velankar SS, Badylak SF. A hydrogel derived from decellularized dermal extracellular matrix. Biomaterials. 2012 Oct;33(29):7028-38.
Point 4: If possible, essential experiments as cytotoxicity and cytocompatibility need to be complement.
Response 4:
Thanks for the comment. In this study, we aimed to develop a promising procedure to prepare ACM hydrogel derived from cartilage. Different parameters were compared and optimized to produce the ideal ACM hydrogel. After the properties of ACM hydrogel were characterized in this study, indeed the next step we plan to conduct cell experiments and in vivo animal tests for verifying the safety and functions of this hydrogel (section 4, line 455-458). From our preliminary cell study, the developed ACMH exerted no cytotoxicity on 3T3 cells. However, more experiments need to be performed in detail in the near future.

Reviewer 3 Report
The manuscript entitled “Preparation and characterization of extracellular matrix hydrogels derived from acellular cartilage tissue” is an interesting manuscript. The authors did detailed experiments to develop acellular cartilage matrix (ACM) hydrogels with minimal adverse effects on the extracellular matrix (ECM). It is a good organised study and ACM hydrogels have a promising potential to be an alternative scaffold for cartilage defect treatment.
Author Response
Response to Reviewer 3 Comments
Dear editor and reviewer,
Thank you for the review of the manuscript entitled "Preparation and characterization of extracellular matrix hydrogels derived from acellular cartilage tissue" (jfb-2053759). We thank the constructive comments and have revised our manuscript accordingly in the text. The responses are highlighted in red font and listed below. Thank you for the prompt attention.
Yours sincerely,
Professor Hsu-Wei Fang, Ph.D. (Corresponding author)
Department of Chemical Engineering and Biotechnology, National Taipei University of Technology. Taipei 10608, Taiwan. (hwfang@ntut.edu.tw)
Reviewer 3 Comments
The manuscript entitled “Preparation and characterization of extracellular matrix hydrogels derived from acellular cartilage tissue” is an interesting manuscript. The authors did detailed experiments to develop acellular cartilage matrix (ACM) hydrogels with minimal adverse effects on the extracellular matrix (ECM). It is a good organised study and ACM hydrogels have a promising potential to be an alternative scaffold for cartilage defect treatment.
Response :
Thanks for the comments!

Reviewer 4 Report
In general, a well-planned and prepared article dedicated to preparation and characterization of dECM hydrogels for cartilage tissue engineering.
However, in my opinion, there is not enough data to accept this work for publication.
Main critical comments:
1) What about the sterilization of this material? Since the authors initially aimed at creating a biomaterial for cartilage tissue engineering, it is necessary to understand how this biomaterial will be sterilized and whether it can be sterilized without loss of its biological properties.
2) Tests to prove the biocompatibility of this material must be done. At least in vitro cell culture tests, live/dead assays, etc. Ideally, subcutaneous implantation, followed by histological analysis.
3) Question to the composition. The authors state that the material consists only of type II collagen and GAG. The given percentages are incomprehensible: how can there be 61.18% GAG and 78.29% collagen in the same material? No more components? I strongly doubt that such a well-defined composition is achievable for dECM.
4) The possible immunogenicity of this material raises concerns. If it is not possible to evaluate it, then it should at least be discussed in detail.
General comments
1) Rheology. I would recommend to add the measurement of frequency sweep tests of both modulus (storage modulus (G') and loss modulus (G''))
2) Characterization of dECM. In the results stated that the amount of collagen type II was measured, however, the test that was used does not allow differentiating between collagens of different types. In fact, the total amount of collagen is measured. At least proof that this material is free of type I collagen is required. Without this, it cannot be argued that this preparation contains only type II collagen.
Author Response
Response to Reviewer 4 Comments
Dear editor and reviewer,
Thank you for the review of the manuscript entitled "Preparation and characterization of extracellular matrix hydrogels derived from acellular cartilage tissue" (jfb-2053759). We thank the constructive comments and have revised our manuscript accordingly in the text. The responses are highlighted in red font and listed below. Thank you for the prompt attention.
Yours sincerely,
Professor Hsu-Wei Fang, Ph.D. (Corresponding author)
Department of Chemical Engineering and Biotechnology, National Taipei University of Technology. Taipei 10608, Taiwan. (hwfang@ntut.edu.tw)
Reviewer 4 Comments
In general, a well-planned and prepared article dedicated to preparation and characterization of dECM hydrogels for cartilage tissue engineering.
However, in my opinion, there is not enough data to accept this work for publication.
Main critical comments:
Point 1: What about the sterilization of this material? Since the authors initially aimed at creating a biomaterial for cartilage tissue engineering, it is necessary to understand how this biomaterial will be sterilized and whether it can be sterilized without loss of its biological properties.
Response 1:
Although sterilization or disinfection may affect physic-chemical properties of biomaterials, they need to be in the aseptic state prior to implantation or in vitro use. The most common sterilization methods are irradiation and ethylene oxide (EO). Currently most commercialized tissue-derived extracellular matrix such as bovine collagen matrix products are sterilized by gamma radiation because it is simple with no residual toxicity. In a review paper, the authors recommended that EO is better for materials with organic substances retained (e.g. protein or polysaccharide) (Tao et al., 2021)
Reference:
Tao M, Ao T, Mao X, Yan X, Javed R, Hou W, Wang Y, Sun C, Lin S, Yu T, Ao Q. Sterilization and disinfection methods for decellularized matrix materials: Review, consideration and proposal. Bioact Mater. 2021 Feb 27;6(9):2927-2945.
Point 2: Tests to prove the biocompatibility of this material must be done. At least in vitro cell culture tests, live/dead assays, etc. Ideally, subcutaneous implantation, followed by histological analysis.
Response 2:
Thanks for the comment. In this study, we aimed to develop a promising procedure to prepare ACM hydrogel derived from cartilage. Different parameters were compared and optimized to produce the ideal ACM hydrogel. After the properties of ACM hydrogel were characterized in this study, indeed the next step we plan to conduct cell experiments and in vivo animal tests for verifying the safety and functions of this hydrogel(section 4, line 455-458). From our preliminary cell study, the developed ACMH exerted no cytotoxicity on 3T3 cells. However, more experiments need to be performed in detail in the near future.
Point 3: Question to the composition. The authors state that the material consists only of type II collagen and GAG. The given percentages are incomprehensible: how can there be 61.18% GAG and 78.29% collagen in the same material? No more components? I strongly doubt that such a well-defined composition is achievable for dECM.
Response 3:
The GAG (61.18 %) and collagen (78.29 %) contents were compared with control hydrogel prepared without decellularized procedure (100 %). The percentage does not mean its relative amount in this hydrogel. To avoid misunderstanding, we have modified the sentences in the manuscript and make it more clear.
Point 4: The possible immunogenicity of this material raises concerns. If it is not possible to evaluate it, then it should at least be discussed in detail.
Response 4:
Thanks for the comment. We have modified the discussion section and addressed more about the issue of immunogenicity. Please see section 4, line 331-369:
“But cells or antigens present in xenogeneic or allogeneic materials may trigger adverse immune responses. Therefore, decellularization aims to remove xenogeneic nucleic acids, antigens, etc., to avoid adverse severe immune responses [22]. The main factors inducing severe immunity are cell surface antigens and nucleic acid substances. Researchers have used physical, chemical, and enzyme methods to remove cell-related substances and to retain ECM components and structures to prepare decellularized matrix materials [23].
Compared with other tissues, the structure of cartilage tissue is denser, making it more difficult to remove cellular immunogens and retain natural matrix components [24]. Kridel et al. [25] used artificial dermis to repair the perforation of nasal septal cartilage in patients. After three months of observation, it was found that the repairment was similar to that of autologous tissue. Porzionato et al. [26] employed alcohol dehydration, freeze-drying, and repeated freezing and thawing to decellularize rabbit nasal septal cartilage and then implanted chondrocytes as scaffolds. However, because cartilage tissue still retained considerable antigenicity, implanting it into living bodies was prone to adverse immune responses. Thus, Narez et al. [27] used non-ionic surfactants to treat meniscal cartilage tissue. Although it could maintain structural integrity, it could not entirely remove the cells.
Chemical and enzymatic methods were used to prepare decellularization solution, combined with physical methods, frozen cartilage powder, and ultrasonic shock to remove cells. Regarding the surface antigen, it has been pointed out that violent hyperacute rejection is elicited when native or incompletely decellularized porcine heart valves are transplanted into humans, similar to the reactions in ABO incompatibility mainly because of the surface residues alpha-gal epitope (galactose-alpha1–3-galactose) on the porcine cell [28]. However, a few studies have pointed out that allograft cartilage does not induce apparent immune rejection [29, 30]. Moreover, Revell and Athanasiou [31] showed that a large amount of cartilage matrix could reduce the contact area between the host T-cell and the antigen of the source chondrocyte so that the immune response is reduced or does not occur. Therefore, the success rate of tissue repair can be improved. The in vivo findings of Huey et al. [32] pointed out that, as long as the alpha-gal epitope can be successfully removed in the cartilage transplantation experiment, no other immune rejection will occur.
The threshold concentration of residual cellular material within ECM sufficient to elicit adverse host responses has been suggested based on some in vivo studies that adverse cell and host immune reactions have been avoided. For example, dsDNA per mg ECM dry weigh should be less than 50 ng, or there should be lack of visible nuclear material in tissue sections [33]. The ACM and ACM hydrogel produced in this study fit the criteria that after the cell removal procedure, most cells and DNA were removed from the cartilage matrices of all three sizes. (Fig. 2, Fig. 3a and Fig. 6a). “
General comments
Point 5: Rheology. I would recommend to add the measurement of frequency sweep tests of both modulus (storage modulus (G') and loss modulus (G''))
Response 5:
Rheology is typically utilized to determine the storage modulus, or stiffness, of the hydrogel following gelation. By rheological measurement, the loss modulus was relatively low as base line in our study. From an article regarding hydrogel derived from decellularized dermal extracellular matrix, similar result was also observed and only storage modulus was shown (Wolf et al., 2012).
Reference:
Wolf MT, Daly KA, Brennan-Pierce EP, Johnson SA, Carruthers CA, D'Amore A, Nagarkar SP, Velankar SS, Badylak SF. A hydrogel derived from decellularized dermal extracellular matrix. Biomaterials. 2012 Oct;33(29):7028-38.
Point 6: Characterization of dECM. In the results stated that the amount of collagen type II was measured, however, the test that was used does not allow differentiating between collagens of different types. In fact, the total amount of collagen is measured. At least proof that this material is free of type I collagen is required. Without this, it cannot be argued that this preparation contains only type II collagen.
Response 6:
Thanks for the reminder. Indeed we did not examine the amount of collagen II and the sentence was not proper in the manuscript. We have removed “type II” in the article and only showed that its a measurement of collagens in the samples.

Round 2
Reviewer 4 Report
Point 1: What about the sterilization of this material? Since the authors initially aimed at creating a biomaterial for cartilage tissue engineering, it is necessary to understand how this biomaterial will be sterilized and whether it can be sterilized without loss of its biological properties.
Response 1:
Although sterilization or disinfection may affect physic-chemical properties of biomaterials, they need to be in the aseptic state prior to implantation or in vitro use. The most common sterilization methods are irradiation and ethylene oxide (EO). Currently most commercialized tissue-derived extracellular matrix such as bovine collagen matrix products are sterilized by gamma radiation because it is simple with no residual toxicity. In a review paper, the authors recommended that EO is better for materials with organic substances retained (e.g. protein or polysaccharide) (Tao et al., 2021)
Reference:
Tao M, Ao T, Mao X, Yan X, Javed R, Hou W, Wang Y, Sun C, Lin S, Yu T, Ao Q. Sterilization and disinfection methods for decellularized matrix materials: Review, consideration and proposal. Bioact Mater. 2021 Feb 27;6(9):2927-2945.
Comment for Response 1:
Thanks for the helpful link, but that doesn't answer my main question. The mentioned goal of your paper is «To designe and characterize ACM hydrogels (line 83)». Sterilization of the dECM is important for both in vitro and in vivo use. Sterilization is a main part of the biomaterial production/designing process. Without a validated sterilization process of your material can’t be used as biomaterial. And as it was already mentioned above sterilization can affect physical, mechanical and biological properties of the biomaterial. Therefore, the study of these properties must be carried out after sterilization process. So, sterilization stage must be added to the methodological part of your paper.
As for EO sterilization of hydrogels – this method is minimally effective in hydrogels and it can affect the mechanical properties of the final hydrogel. I recommend for you to read this review –
McInnes, A.D.; Moser, M.A.J.; Chen, X. Preparation and Use of Decellularized Extracellular Matrix for Tissue Engineering. J. Funct.Biomater.2022,13, 240. https://doi.org/10.3390/jfb13040240
Point 2: Tests to prove the biocompatibility of this material must be done. At least in vitro cell culture tests, live/dead assays, etc. Ideally, subcutaneous implantation, followed by histological analysis.
Response 2:
Thanks for the comment. In this study, we aimed to develop a promising procedure to prepare ACM hydrogel derived from cartilage. Different parameters were compared and optimized to produce the ideal ACM hydrogel. After the properties of ACM hydrogel were characterized in this study, indeed the next step we plan to conduct cell experiments and in vivo animal tests for verifying the safety and functions of this hydrogel(section 4, line 455-458). From our preliminary cell study, the developed ACMH exerted no cytotoxicity on 3T3 cells. However, more experiments need to be performed in detail in the near future.
Comment for Response 2:
Without in vitro cell test and in vivo experiment it is prematurely draw conclusions about «ideality» of ACM hydrogel. Although I am willing to accept that you can transfer these works to the near future. But I advise you to use primary cells for in vitro tests, and do not use 3T3 cells.
Other questions is fine. I haven’t got issues with them.
Author Response
Response to Reviewer 4 Comments (round 2)
Dear editor and reviewer,
Thank you for the second review of the manuscript entitled "Preparation and characterization of extracellular matrix hydrogels derived from acellular cartilage tissue" (jfb-2053759). We thank the constructive comments which notably enhance the quality of this manuscript and we have revised our manuscript accordingly in the text. The responses are highlighted in red font and listed below. Thank you for the prompt attention.
Yours sincerely,
Professor Hsu-Wei Fang, Ph.D. (Corresponding author)
Department of Chemical Engineering and Biotechnology, National Taipei University of Technology. Taipei 10608, Taiwan. (hwfang@ntut.edu.tw)
Reviewer 4 Comments (round 2)
Point 1: What about the sterilization of this material? Since the authors initially aimed at creating a biomaterial for cartilage tissue engineering, it is necessary to understand how this biomaterial will be sterilized and whether it can be sterilized without loss of its biological properties.
Response 1:
Although sterilization or disinfection may affect physic-chemical properties of biomaterials, they need to be in the aseptic state prior to implantation or in vitro use. The most common sterilization methods are irradiation and ethylene oxide (EO). Currently most commercialized tissue-derived extracellular matrix such as bovine collagen matrix products are sterilized by gamma radiation because it is simple with no residual toxicity. In a review paper, the authors recommended that EO is better for materials with organic substances retained (e.g. protein or polysaccharide) (Tao et al., 2021)
Reference:
Tao M, Ao T, Mao X, Yan X, Javed R, Hou W, Wang Y, Sun C, Lin S, Yu T, Ao Q. Sterilization and disinfection methods for decellularized matrix materials: Review, consideration and proposal. Bioact Mater. 2021 Feb 27;6(9):2927-2945.
Comment for Response 1:
Thanks for the helpful link, but that doesn't answer my main question. The mentioned goal of your paper is «To designe and characterize ACM hydrogels (line 83)». Sterilization of the dECM is important for both in vitro and in vivo use. Sterilization is a main part of the biomaterial production/designing process. Without a validated sterilization process of your material can’t be used as biomaterial. And as it was already mentioned above sterilization can affect physical, mechanical and biological properties of the biomaterial. Therefore, the study of these properties must be carried out after sterilization process. So, sterilization stage must be added to the methodological part of your paper.
As for EO sterilization of hydrogels – this method is minimally effective in hydrogels and it can affect the mechanical properties of the final hydrogel. I recommend for you to read this review –
McInnes, A.D.; Moser, M.A.J.; Chen, X. Preparation and Use of Decellularized Extracellular Matrix for Tissue Engineering. J. Funct.Biomater.2022,13, 240. https://doi.org/10.3390/jfb13040240
Response 1 for round 2:
Thanks for the reminder and this informative/detailed article. In section 2.2 line (120-124), we actually mentioned that “Following the removal of the cells, the ACM was centrifuged at 2000 rpm for 10 minutes, shook, and washed with ddH2O for 15 minutes until no foam was visible. After extracting the supernatant, 10 mL of ddH2O was added and the tubes were frozen at -80°C for 5 hours, freeze-dried for 48 hours, exposed to UV irradiation for 24 hours, and placed in a drying oven.” After sterilizing the ACM powder by UV, we then prepared the hydrogel with other sterilized reagents and performed the following physical, chemical and ex vivo examinations.
To address the sterilization issue which could be very important for the properties of hydrogel, we have added some discussion in section 4 and cited the the above paper from McInnes et al. as a reference. Please check Section 4, line 389-393: “It is worth noting that the sterilization of the biomaterials would be a critical issue that it could change the biological or mechanical properties of the materials. There have been many techniques proposed to disinfect biomaterials including peracetic acid and/or ethanol, antimicrobials, UV, ethylene oxide, and gamma radiation...etc. In this study, ACM was disinfected by UV exposure and mixed with sterilized reagents to obtain ACMH for further examinations.”
Point 2: Tests to prove the biocompatibility of this material must be done. At least in vitro cell culture tests, live/dead assays, etc. Ideally, subcutaneous implantation, followed by histological analysis.
Response 2:
Thanks for the comment. In this study, we aimed to develop a promising procedure to prepare ACM hydrogel derived from cartilage. Different parameters were compared and optimized to produce the ideal ACM hydrogel. After the properties of ACM hydrogel were characterized in this study, indeed the next step we plan to conduct cell experiments and in vivo animal tests for verifying the safety and functions of this hydrogel(section 4, line 455-458). From our preliminary cell study, the developed ACMH exerted no cytotoxicity on 3T3 cells. However, more experiments need to be performed in detail in the near future.
Comment for Response 2:
Without in vitro cell test and in vivo experiment it is prematurely draw conclusions about «ideality» of ACM hydrogel. Although I am willing to accept that you can transfer these works to the near future. But I advise you to use primary cells for in vitro tests, and do not use 3T3 cells.
Other questions is fine. I haven’t got issues with them.
Response 2 for round 2:
Thanks for the suggestion. Indeed primary cells would be more suitable and provide more information to investigate the ACM hydrogel developed in this study. We will use primary chondrocytes from human source as in vitro model for the study of near future. We modified our sentences in section 4 discussion, line 460-464: “The cytotoxicity and immunogenicity of the cartilage-derived hydrogel will be also evaluated to address the safety issue of this material by testing with human primary chondrocytes. The future works will focus on the cartilage repair effects by testing ACM hydrogel with human primary chondrocytes and cartilage defect animal models.”

Round 3
Reviewer 4 Report
I haven't got any questions more